



# Global total ozone recovery trends derived from five merged ozone datasets

Mark Weber[1], Carlo Arosio[1], Melanie Coldewey-Egbers[2], Vitali E. Fioletov[3], Stacey M. Frith[4], Jeannette D. Wild[5,6], Kleareti Tourpali[7], John P. Burrows[1], and Diego Loyola[2]

[1]University of Bremen, Bremen, Germany
[2]German Aerospace Center (DLR), Oberpfaffenhofen, Germany
[3]Environment and Climate Change Canada, Toronto, Canada
[4]Science Systems and Applications Inc., Lanham, MD, USA
[5]NOAA/NCEP Climate Prediction Center, College Park, MD, USA
[6]CISESS/ESSIC, UMD, College Park, MD, USA
[7]Aristotle University, Thessaloniki, Greece

**Correspondence:** Mark Weber (weber@uni-bremen.de)

**Abstract.** We report on updated trends using different merged zonal mean total ozone datasets from satellite and ground-based observations for the period from 1979 to 2020. This work is an update from the trends reported in Weber et al. (2018) using the same datasets up to 2016. Merged datasets used in this study include NASA MOD v8.7 and NOAA Cohesive Data (COH) v8.6, both based on data from the series of Solar Backscatter UltraViolet (SBUV), SBUV-2, and Ozone Mapping and Profiler Suite (OMPS) satellite instruments (1978–present) as well as the Global Ozone Monitoring Experiment (GOME)-type Total Ozone (GTO-ECV) and GOME-SCIAMACHY-GOME-2 (GSG) merged datasets (both 1995–present), mainly comprising satellite data from GOME, SCIAMACHY, OMI, GOME-2A, -2B, and TROPOMI. The fifth dataset consists of the annual mean zonal mean data from ground-based measurements collected at the World Ozone and UV Radiation Data Center (WOUDC).

Trends were determined by applying a multiple linear regression (MLR) to annual mean zonal mean data. The addition of four more years consolidated the fact that total ozone is indeed on slowly recovering in both hemispheres as a result of phasing out ozone depleting substances (ODS) as mandated by the Montreal Protocol. The near global ozone trend of the median of all datasets after 1996 was $0.5 \pm 0.2$ $(2\sigma)$ %/decade, which is in absolute numbers roughly a third of the decreasing rate of $1.4 \pm 0.6$ %/decade from 1978 until 1996. The ratio of decline and increase is nearly identical to that of the EESC (equivalent effective stratospheric chlorine or stratospheric halogen) change rates before and after 1996 which confirms the success of the Montreal Protocol. The observed trends are also in very good agreement with the median of 17 chemistry climate models from CCMI (Chemistry Climate Model Initiative) with current ODS and GHG (greenhouse gas) scenarios.

The positive ODS related trends in the NH after 1996 are only obtained with a sufficient number of terms in the MLR accounting properly for dynamical ozone changes (Brewer-Dobson circulation, AO, AAO). A standard MLR (limited to solar, QBO, volcanic, and ENSO) leads to zero trends showing that the small positive ODS related trends have been balanced by negative trend contributions from atmospheric dynamics resulting in nearly constant total ozone levels since 2000.





# 1 Introduction

The stratospheric ozone layer protects the biosphere from harmful UV radiation. How much UV reaches the surface depends, among other factors like clouds, on the overhead total ozone column. The discovery of the Antarctic ozone hole (Chubachi, 1984; Farman et al., 1985; Solomon et al., 1986) raised the awareness of the need to protect the ozone layer that culminated in the 1985 Vienna Convention and a commitment to take actions. One of the actions was the signing of the Montreal Protocol in 1987 that started the phaseout of ozone depleting substances (ODS), which are sufficiently long-lived to reach the stratosphere and release active halogens that destroy ozone (e.g. Solomon, 1999). As a consequence of the Montreal Protocol and its later amendments stratospheric halogens started to decline in the middle 1990s (e.g. Anderson et al., 2000; Solomon et al., 2006). A corresponding ozone increase has been detected from satellite and ground-based observations, particularly in the upper stratopshere (Braesicke et al., 2018, and references therein).

Changes in total ozone column are representative of lower stratospheric ozone changes as the majority of ozone resides in the lower stratosphere ("ozone layer"). Lower stratospheric ozone is sufficiently long-lived to be influenced by transport and circulation changes. The rapid increase in northern hemisphere total ozone in the late 1990s (Harris et al., 2008) revealed the important role of ozone transport via the Brewer-Dobson (BD) circulation. These circulation changes also cause large variability on inter- and intra-annual time scales in lower stratospheric ozone and the total column (e.g. Fusco and Salby, 1999; Randel et al., 2002; Dhomse et al., 2006; Harris et al., 2008; Weber et al., 2011) and make detection of ozone recovery challenging. Apart from the observed variability, zonal mean total ozone levels in both hemispheres remained stable since about the year 2000 (e.g. Weber et al., 2018). The success of the Montreal Protocol agreement is nevertheless undisputed as the earlier decline in total ozone was successfully stopped (Mäder et al., 2010; Braesicke et al., 2018).

Global and continuous ozone observations from satellites through 2020 now span a total time period of forty-two years, of which 25 years cover the period after the stratospheric halogen peak (around 1996). The added years should help in improving the statistical significance of ozone recovery after the middle 1990s (Weatherhead et al., 2000). This paper reports on updated zonal mean total ozone trends from Weber et al. (2018) (abbreviated to W18 in the following) by adding four more years of data (2017-2020) to five merged total ozone datasets. In our earlier study ozone recovery trends in the extratropics were on the order of 0.5 %/decade. The derived trends depend on the proper treatment of dynamical processes in the multi-linear regression. Changes in circulation and ozone transport, in part due to increasing greenhouse gas levels (GHG), have variability on decadal and longer time scales and can therefore mask ODS related recovery trends. Longer data records are helpful to further disentangle the various processes responsible for long-term changes in ozone.

The main results from our earlier paper (W18) were latitude dependent annual mean total ozone trends from the middle 1990s to 2016, which were reported to be on average +0.5 %/decade in the extratropics and only significant in the SH (W18). Since W18 was published there were three recent studies on global and regional ozone column trends (Bozhkova et al., 2019; Krzyścin and Baranowski, 2019; Coldewey-Egbers et al., 2021). Krzyścin and Baranowski (2019) derived total ozone column trends from a multivariate linear regression (MLR) applied to the Multi-Sensor Reanalysis-2 (MSR-2) total ozone dataset up to 2017 (van der A et al., 2015). In their MLR they split the entire period from 1978 to 2017 into three periods with separate



trends (either independent or piecewise linear). The choice of two inflection points were chosen from fits having minimum fit
root mean square (rms) errors. As stratospheric halogens are declining steadily since the middle 1990s the interpretation of the
segmented trends is difficult. Trends of the first period (before middle 1990s) are in agreement with W18 and this study.

Bozhkova et al. (2019) applied a regression to TOMS and OMI total ozone at northern hemispheric mid-latitudes using the
approach by Bloomer et al. (2010), first applied to surface ozone and temperature data at selected stations in the US. Without
using any proxy data the regression estimates trends of the seasonality expressed as Fourier series. Attribution of physical and
chemical processes to the long-term changes are therefore not possible as also stated by the authors. Latitude and longitude
dependent total ozone trends are reported by Coldewey-Egbers et al. (2021) derived from the ESA/DLR GTO-ECV dataset,
which is one of the five observational datasets used in this study. They report significant positive linear trends after 1995 over
64 large regions in the extratropical southern hemisphere, while in the tropics and NH they are mostly insignificant. Consequently,
they only reported significant zonal mean positive trends in the SH.

In Section 2 the updates in the five merged datasets are briefly discussed. In Section 3 the multiple linear regression (MLR)
as used in our trend analysis is described. Section 4 presents the total ozone trend results in broad zonal bands: near-global,
southern and northern hemispheric extratropics, and tropics. In Section 5 latitude dependent annual mean total ozone trends
are presented and discussed. Polar ozone trends for the months where polar ozone losses are largest (e.g. during ozone hole
season) are presented in Section 6. In Section 7 a summary and final remarks are given.

## 2 Total ozone datasets

Five merged total ozone datasets are used in this study of which one dataset is based upon ground observations. All others are
based on satellite observations. Two different merged datasets are derived from the series of SBUV and SBUV-2 satellite instru-
74 ments (SBUV MOD V8.7 from NASA and SBUV COH V8.6 from NOAA) operating continuously since the late 1970s. The
other two merged datasets are based in large part upon the series of European satellite spectrometers GOME, SCIAMACHY,
GOME-2A, and GOME-2B with different retrieval and merging algorithms applied (University of Bremen GSG and ESA/DLR
GTO-ECV datasets). These datasets start in 1995.

The ground based dataset is the monthly mean zonal mean data from the network of ground-based Brewers, Dobsons, SAOZ
(Système d'Analyse par Observations Zénithales), and filter instruments collected at the World Ozone and UV Data Center
(WOUDC) (Fioletov et al., 2002). In addition a brief description of the model data from the CCMI initiative is given. The
sources of observational data are listed in Table 1 and brief descriptions of the datasets are given in the following. Annual mean
timeseries of all five merged datasets are in very good agreement with each other (see Fig. 2.58 in Weber et al. (2021)).

### 2.1 NASA SBUV MOD V8.7

The NASA Merged Ozone Data (MOD) time series is constructed using data from the Nimbus 4 BUV, Nimbus 7 SBUV,
and six NOAA SBUV-2 instruments numbered 11, 14, and 16-19, and the Ozone Mapping and Profiler Suite Nadir Profiler
(OMPS-NP) instrument aboard the Suomi-NPP satellite (Frith et al., 2014, 2022).The instruments are of similar design, and





**Table 1.** Source of merged total ozone datasets.

| Dataset | Start year | Source |
|---------|-----------|--------|
| NASA SBUV MOD V8.7 | 1970 | http://acdb-ext.gsfc.nasa.gov/Data_services/merged/ |
| NOAA SBUV COH V8.6 | 1978 | ftp://ftp.cpc.ncep.noaa.gov/SBUV_CDR/ |
| GSG | 1995 | http://www.iup.uni-bremen.de/gome/wfdoas |
| GTO | 1995 | http://atmos.eoc.dlr.de/gome/gto-ecv.html |
| WOUDC | 1964 | http://woudc.org/archive/Projects-Campaigns/ZonalMeans/ |

measurements from each are processed using the same V8.7 retrieval algorithm. To maintain consistency over the entire time
series the individual instrument records are analyzed with respect to each other and absolute calibration adjustments are applied
as needed based on comparison of radiance measurements during periods of instrument overlap (DeLand et al., 2012).

Version 8.7 uses the same core algorithm as Version 8.6 (Bhartia et al., 2013) but includes new inter-instrument calibration
adjustments for instrument records since 2000 (NOAA-16 SBUV/2 though OMPS NP) based on a new approach to radiance
intercomparisons across overlapping instruments (Kramarova et al., 2022). Version 8.7 also incorporates an updated a-priori
with improved tropospheric representation based on GMI model output, and diurnal adjustments to ensure the a-priori profile
correctly reflects the local solar time of each measurement (Ziemke et al., 2021). A post-retrieval diurnal correction is applied
to adjust each instrument record to an equivalent measurement time of 1:30pm (Frith et al., 2020). Remaining offsets between
instruments exist (mostly below 5% for layers, below 1% for total ozone), but their cause is not understood. We therefore do
not make adjustments to the data. Rather we set limitations on the data included in the merged product based on data quality
analysis by the instrument team and on comparisons with independent measurements (DeLand et al., 2012; Kramarova et al.,
2013, 2022). For merging, data are averaged during periods with multiple operational instruments. The Version 8.7 MOD data
contains monthly zonal mean ozone profiles in mixing ratio on pressure levels and in Dobson units on layers. The total ozone
is then provided as the sum of the layer data.

**2.2 NOAA SBUV COH V8.6**

The NOAA COH (cohesive) dataset is a simple extension in time of the dataset appearing in W18. The data includes v8.6 SBUV
on Nimbus 7, v8.6 SBUV/2 from NOAA 9, 11, 16 to 19, and v2r3 OMPS Nadir Profiler (NP) on Suomi-NPP as available from
NESDIS STAR. The merging approach differs from NASA MOD in two important ways. NASA MOD averages data from
all relevant satellites in any time period for which the data meets certain quality criteria. NOAA COH uses data from a single
'best' satellite in any time period. Which satellite is used depends on known data quality issues, on minimizing the solar zenith
angle of the measurement, and on maximizing global coverage. NOAA COH does not shift to a equivalent measurement time
(1:30pm), but performs an adjustment between data from differing satellites. For post 2000 data, where drift of the measurement
time is minimized, the data are all adjusted to NOAA 18. For data 1999 and prior, the inter-satellite overlap is often short, the
satellite drift often significant, we choose only to adjust NOAA 9 to the two branches NOAA 11 prior and after the NOAA 9





time period. The total ozone is calculated from the sum of the adjusted profile layer data. By vertical integration many of the layer adjustments to a large extent cancel such that the final total ozone product is altered by less than 1%, and in most cases

by less than 0.5%, from the original satellite datasets.

### 2.3  University of Bremen GSG

The merged GOME, SCIAMACHY, GOME-2A and -2B (GSG) total ozone timeseries (Kiesewetter et al., 2010; Weber et al., 2011, 2018) consists of total ozone data that were retrieved using the University of Bremen Weighting Function DOAS (WF-

DOAS) algorithm (Coldewey-Egbers et al., 2005; Weber et al., 2005; Orfanoz-Cheuquelaf et al., 2021). The merging of the data has been described in W18. The most recent modification was to replace GOME-2A data after January 2015 with data

from GOME-2B (2012-present) which has a better global coverage after changes in the GOME-2A scanning pattern. Latitude dependent bias corrections for GOME-2B were applied from the overlapping period 2014-2020 with GOME-2A.

### 2.4  DLR/ESA GTO-ECV

The latest version of the GOME-type Total Ozone Essential Climate Variable (GTO-ECV) data record (Coldewey-Egbers

et al., 2015, 2021; Garane et al., 2018) has been generated as part of the European Space Agency's Climate Change Initiative+ ozone (ESA_CCI+ ozone) project. Total columns from six sensors (GOME, SCIAMACHY, OMI, GOME-2A, GOME-2B, and

TROPOMI), retrieved with the GOME Direct Fitting (GODFIT) version 4 algorithm (Lerot et al., 2014; Garane et al., 2018), were combined into a coherent record that covers the period 1995-2020. OMI was used as a reference instrument and the other

sensors were adjusted by means of latitude and time dependent correction factors determined from overlap periods.

### 2.5  WOUDC data

The WOUDC ground-based zonal mean data set (Fioletov et al., 2002) was formed from ground-based measurements by Dobson, Brewer, SAOZ instruments, and filter ozonometers available from the WOUDC. The overall performance of the

ground-based network was discussed by Fioletov et al. (2008).

First, ground-based measurements were compared with an ozone "climatology" (monthly means for each point of the globe)

estimated from satellite data for 1978–1989. Then, for each station and for each month the deviations from the climatology were calculated, and the belt's value for a particular month was estimated as a mean of these deviations. The calculations were

done for 5° latitudinal belts.

In order to take into account various densities of the network across regions, the deviations of the stations were first averaged

over 5° by 30° cells, and then the belt mean was calculated by averaging these first set of averages over the belts. Then the zonal averages were smoothed by approximating them using Legendre polynomials.

The WOUDC data set was compared with merged satellite time series and demonstrated a good agreement (Chiou et al., 2014). Estimates based on relatively sparse ground-based measurements, particularly in the tropics and southern hemisphere,





may not always reproduce monthly zonal mean fluctuations well. However, seasonal (and longer) averages can be estimated with a precision comparable with satellite-based data sets ($\sim$1%).

## 2.6 Chemistry climate model data

In this study output from the chemistry–climate models (CCMs) and chemistry-transport models (CTMs) participating in phase 1 of CCMI (Chemistry-Climate Model Initiative) are used (Eyring et al., 2013). An overview of the models, together with details particular to each model and an overview of the available simulations, is given in Morgenstern et al. (2017) along with a detailed description of the full forcings used in the reference simulations (Eyring et al., 2013; Hegglin et al., 2016). Here we have used median total column ozone from 17 models taking part in the REF-C2 experiment, an internally consistent seamless simulation from the past into the future between 1960 and 2100.

## 2.7 Data preparation

From the zonal mean monthly mean data in 5° latitude steps (all datasets) annual means were calculated. Wider zonal bands (like 35°N-60°N) were averaged from the 5° data using area weights (see W18). All annual mean zonal mean timeseries were bias corrected by subtracting the difference to the mean of all datasets during the 1998-2008 period. The multi-dataset mean was then added back to each dataset, such that all bias corrected timeseries are provided in units of the total column amounts (W18). However, the trend results derived from them are identical to those derived using anomaly timeseries.

Like in our earlier study, the GSG and GTO-ECV timeseries were extended from 1995 back to 1979 using the bias corrected NOAA data. This way one ensures that all terms other than the trend terms are determined from the full time (1979-2020) period. The NOAA data was here preferred over the NASA data, as the former has shorter data gaps after the major volcanic eruption from Mt Pinatubo in 1991 and subsequent years.

## 3 Multiple linear regression

The standard MLR model is identical to the one used in W18 and includes two independent linear trend terms (before and after the ODS related turnaround year $t_0 =$ 1996), two aerosol terms (Mt. Pinatubo 1992 and El Chichón 1983), solar cycle term, two QBO terms (50 and 10 hPa), and ENSO (El Niño-Southern Oscillation):

$$
\begin{aligned}
y(t) \quad &= \big[a_1 + b_1 \cdot (t_0 - t)\big] X_1(t) + \big[a_2 + b_2 \cdot (t - t_0)\big] X_2(t) \\
&+ \alpha_{\text{sun}} \cdot S(t) + \alpha_{\text{qbo50}} \cdot Q_{50}(t) + \alpha_{\text{qbo10}} \cdot Q_{10}(t) + \alpha_{\text{ENSO}} \cdot E(t) \\
&+ \alpha_{\text{ElChichón}} \cdot A_1(t) + \alpha_{\text{Pinatubo}} \cdot A_2(t) + P(t) + \epsilon(t).
\end{aligned}
\tag{1}
$$

$y(t)$ is the annual mean zonal mean total ozone timeseries and $t$ the year of observations. The coefficients $b_1$ and $b_2$ are the linear trends before and after $t_0$. In order to make both trends independent of each other (or disjoint), two y-intercepts ($a_1$ and $a_2$) are added. The multiplication of the independent variable $t$ with $X_i(t)$ in the first four terms of Eq. 1 describes mathematically that the first two terms only applies to the period before and the third and fourth terms to the period after the turnaround year.



$X_1(t)$ and $X_2(t)$ are given by

$$X_1(t) = \begin{cases} 1 & \text{if } t \le t_0 \\ 0 & \text{if } t > t_0 \end{cases} \tag{2}$$

and

$$X_2(t) = \begin{cases} 0 & \text{if } t \le t_0 \\ 1 & \text{if } t > t_0 \end{cases}, \tag{3}$$

respectively. The independent trends before and after $t_0$ are favored over the use of piecewise linear trends or the use of EESC
as a proxy timeseries (see detailed discussions in W18) The maximum of the effective equivalent stratospheric chlorine (EESC)
was reached at about the year $t_0 = 1996$ (Newman et al., 2007) and some years later ($t_0 \sim 2000$) in the polar regions (Newman
et al., 2006, 2007). Therefore $t_0$ was set to 1996 globally, except for the polar regions, where $t_0$=2000 was selected. The
178 contributions from the QBO, 11-year solar cycle, and stratospheric aerosols are standard in total ozone MLR analyses (e.g.
Staehelin et al., 2001; Reinsel et al., 2005). $\epsilon(t)$ is the residual from fitting the coefficients to match the regression model (right
side) to the observations. By using annual mean total ozone, auto-correlation is very low here (below 0.1 in absolute value for
a shift by one year) so that no further additional auto-regression term as commonly used for monthly mean ozone timeseries is
182 needed (e.g. Dhomse et al., 2006; Vyushin et al., 2007).

The stratospheric aerosols are dominated by the major volcanic eruptions from El Chichón (1982) and Mt. Pinatubo (1991).
Enhanced aerosols in the lower stratosphere lasting for a few years impact both ozone chemistry and transport (Schnadt Poberaj
et al., 2011; Dhomse et al., 2015). The stratospheric aerosol optical depth (SAOD) at 550 nm from Sato et al. (1993) is used as
the explanatory variable before 1990 (includes the El Chichón event), while newer data from the WACCM model (Mills et al.,
2016) is used for the period after 1990 (includes Mt. Pinatubo major volcanic eruption and the series of more minor volcanic
eruptions from the last decade). Missing years after 2015 were filled with background values from the late 1990s.

As mentioned in W18 there are not sufficient number of months and/or 5° latitude bands available in the SBUV data records
for some years and thus no annual means were calculated. Annual means were only used in the regression if at least 80% of
the 5° bands of the data were contained in the broad zonal bands and 80% of months available in that year. If annual means of
192 the years 1982 and 1983 are missing, the "El Chichon" term is not used in the MLR, similarly if missing all years from 1991
to 1994, the "Pinatubo" term is excluded in the MLR.

The MLR equation, Eq. 1, without the $P(t)$ term has been commonly applied for determining trends from ozone profile data
(e.g. Bourassa et al., 2014, 2017; Harris et al., 2015; Tummon et al., 2015; Sofieva et al., 2017; Steinbrecht et al., 2017). The
196 extra term $P(t)$ in Eq. 1 accounts for additional factors of dynamical variability that have been used in different combinations
and definitions (e.g. accumulated, time-lagged) in the past. It includes contributions from the Arctic (AO) and Antarctic Os-
198 cilation (AAO), and the Brewer-Dobson circulation (BDC) (e.g. Reinsel et al., 2005; Mäder et al., 2007; Chehade et al., 2014;
Weber et al., 2018). The BDC terms are usually described by the eddy heat flux at 100 hPa that is considered a main driver of
200 the BDC (Fusco and Salby, 1999; Randel et al., 2002; Weber et al., 2011). The term $P(t)$ is given as follows:





**Table 2.** Sources of explanatory variables / proxy timeseries used in the MLR.

| Variable | Proxy | Source |
|---|---|---|
| $S(t)$ | Bremen composite Mg II index (Snow et al., 2014) | http://www.iup.uni-bremen.de/UVSAT/Datasets/mgii |
| $QBO_{50}(t), QBO_{10}(t)$ | Singapore wind speed at 50 and 10 hPa (update from Naujokat, 1986) | http://www.geo.fu-berlin.de/met/ag/strat/produkte/qbo/qbo.dat |
| $E(t)$ | MEI (ENSO) Index (Wolter and Timlin, 2011) | https://www.esrl.noaa.gov/psd/enso/mei/ |
| $AO(t), AAO(t)$ | Antarctic Oscillation (AAO), Arctic Oscillation (AO) | http://www.cpc.ncep.noaa.gov/products/precip/CWlink/daily_ao_index/teleconnections.shtml |
| $A_1(t)$ | stratospheric aerosol depth at 550nm ($t < 1990$) (update from Sato et al., 1993) | https://data.giss.nasa.gov/modelforce/strataer/tau.line_2012.12.txt |
| $A_2(t)$ | stratospheric aerosol depth at 550nm from WACCM model ($t \geq 1990$) (Mills et al., 2016) | http://dx.doi.org/10.5065/D6S180JM |

$$P(t) = \alpha_{\mathrm{AO}} \cdot AO(t) + \alpha_{\mathrm{AAO}} \cdot AAO(t) + \alpha_{\mathrm{BDCn}} \cdot BDCn(t) + \alpha_{\mathrm{BDCs}} \cdot BDCs(t). \tag{4}$$

In W18 the AAO term was not included. Table 2 summarises the sources of the proxy data used here. The calculation of the BDC proxy from the monthly mean eddy heat fluxes is described in detail in W18. In this study the eddy heat flux data come from the ERA-5 reanalysis (Hersbach et al., 2020).

One may argue that the addition of $P(t)$ will lead to some overfitting by the MLR. We justify this addition as it enables us to obtain MLR fits matching the extreme events like very high annual mean ozone in the NH in 2010 and the very large warming events above Antarctica in 2002 and 2019 with unusually high ozone. The better the dynamical variations are represented in the MLR, the more likely we can separate out dynamical trend contributions and the linear trend terms best approximate EESC related trends. In our previous study only selected terms from $P(t)$ were used dependent on their significance in specific zonal bands. Retaining all terms in all MLRs leads to smoother behavior in the latitude dependent ozone response.

The various proxy time series, in particular the atmospheric dynamics related ones, are partially correlated. One way to improve upon this is the possibility to orthogonalize them. Doing so will not change the MLR results, but some contributions from the original proxy terms will be redistributed among the proxies that were orthogonalized. It is also common to detrend the proxy time series. In that case all linear changes of the various processes or proxies will be added up in the linear trend term which makes attribution impossible. The goal here is that linear changes of all the processes as expressed by the various proxy terms shall be excluded from the linear trend terms.



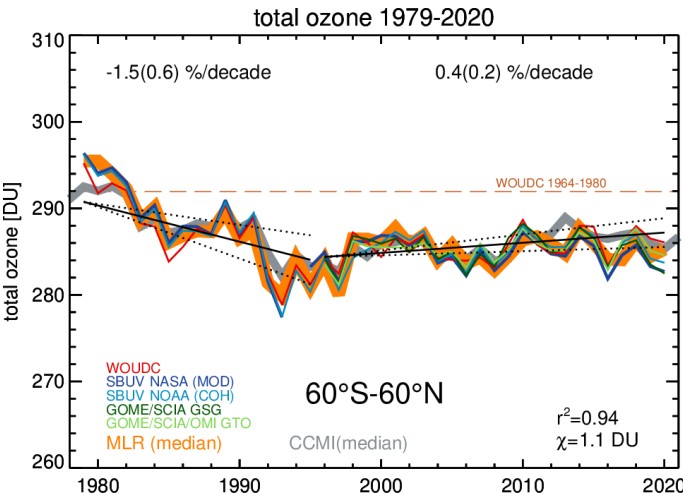

**Figure 1.** Near global (60°S-60°N) total ozone timeseries of five bias corrected merged datasets. The thick orange line is the result from applying the full MLR (Eqs. 1 and 4) to the median timeseries. The square of the correlation between observations and MLR is given by $r^2$. $\chi^2$ is the sum square of the median timeseries minus MLR divided by the degrees of freedom (difference between the number of years, $n$, and number of parameters used in the MLR, $m$). The solid lines indicate the linear trends before and after the ODS peak, respectively. The dotted lines indicate the $2\sigma$ uncertainty of the MLR trend estimates. Trend numbers are indicated for the pre- and post-ODS peak period in the top part of the plot. Numbers in parentheses are the $2\sigma$ trend uncertainty. The orange dashed line shows the mean ozone level from 1964 until 1980 from the WOUDC data. The thick grey line is the median of 17 chemistry-climate models from the CCMI initiative.

## 4   Total ozone trends in broad zonal bands

Figure 1 shows the near-global mean timeseries (60°S-60°N) of the bias-corrected five merged datasets. The thick orange line is the MLR result from applying the full regression model (Eqs. 1 and 4) to the median of the five timeseries. 94% of the variability

in total ozone is well captured by the full MLR. A positive trend of $+0.4\pm0.2\,(2\sigma)$ %/decade after 1996 is derived. This trend is about one third of the absolute trend during the phase of increasing ODS before 1996 which is $-1.5\pm0.6$ %/decade. The

ratio of trends before and after 1996 is very close to the ratio of rate changes in the effective equivalent stratospheric chlorine (EESC) before and after the middle 1990s (Dhomse et al., 2006; Newman et al., 2007). Therefore, the observed linear trend of

roughly half a percent per decade up to 2020 can be interpreted as the recovery from changes in ODS following the Montreal Protocol. This ODS related recovery appears statistically robust (to within $2\sigma$), even though the ozone levels have stayed more

or less constant apart from the year-to-year variability since the year 2000. The magnitude of the post ODS-peak trend remained unchanged from W18. The trend results vary only slightly if the turnaround year (1996) of the ODS change is shifted by one

228   year back and forward. Even if the MLR fit of the post ODS-peak period is limited to years after 2000, the recovery trend remains robust at $+0.5(0.3)$ %/decade.

The current near-global ozone level (2017-2020) is about 2.3% below the average from the 1964-1980 time period, the latter derived from the WOUDC data (see Fig. 1). Recovery of total ozone to the 1980 level is generally not expected before about the



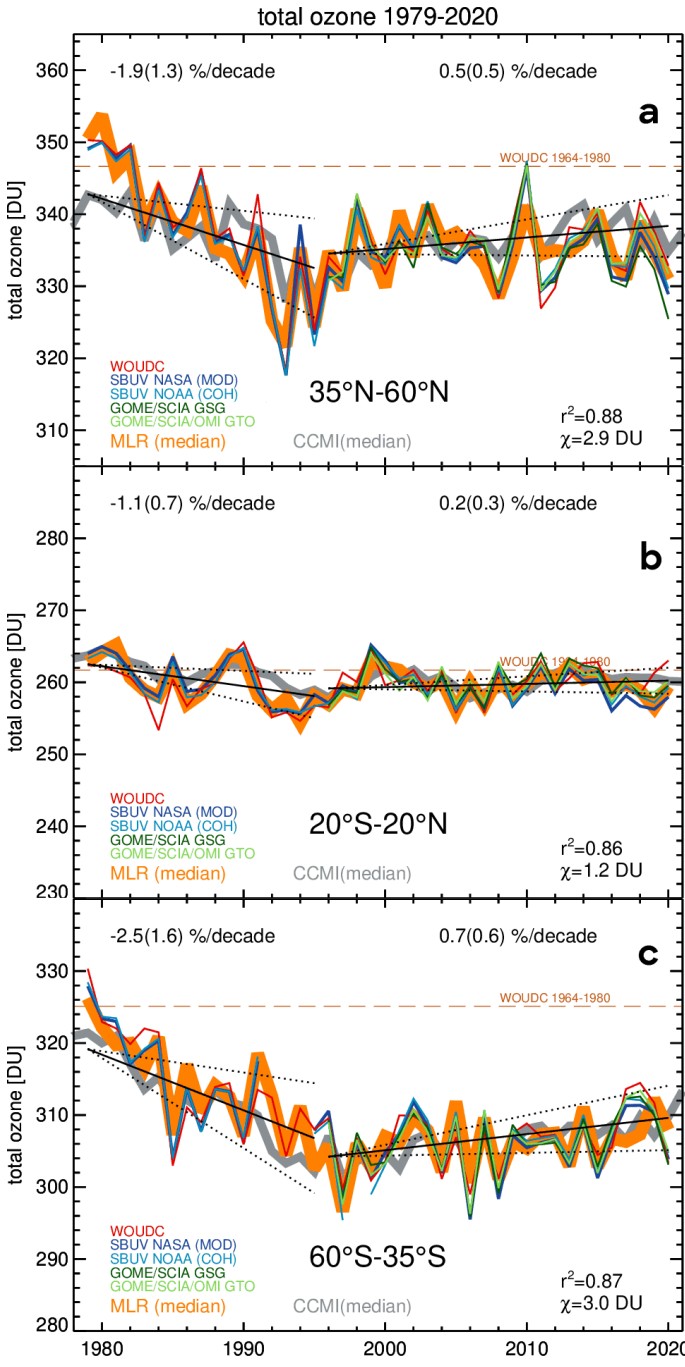

**Figure 2.** Same as Fig. 1, but for broad zonal bands, a) 35°N-60°N (northern hemisphere), b) 20°S-20°N (tropics), and c) 35°S-60°S (southern hemisphere).





middle of this century (Braesicke et al., 2018). The near-global total ozone timeseries from the median of the seventeen CCMI

chemistry-climate models is in very good agreement with the observations from which we conclude that the chemical and

234 dynamical changes in total ozone under current ODS and greenhouse gas (GHG) scenarios are well understood and consistent

with observations.

Figure 2 shows the ozone time series in the northern (NH) and southern hemisphere (SH) as well as in the tropics. Again, the

current ozone levels are well below the 1964-1980 mean, specifically $-3.6\%$ and $-4.7\%$ in the NH and SH ($35°$-$60°$ latitudes),

respectively. The lower value in the SH is due to the influence from the spring Antarctic ozone hole, which exhibits the largest

local ozone depletion and leads to mixing of ozone depleted air into the middle latitudes (Atkinson et al., 1989; Millard et al.,

2002). Recovery trends are $+0.5(0.5)$ and $+0.7(0.6)$ %/decade in the NH and SH, respectively. Within the trend uncertainty,

the 1-to-3 ratio in the linear trends before and after the ODS peak in 1996 are close to the ratio of the rate change in the EESC

in both hemispheres.

In the tropics the linear trend after 1996 is close to zero and insignificant (Fig. 2 and Coldewey-Egbers et al., 2021). Table

3 summarises the MLR results in the broad zonal bands from the individual datasets and the median timeseries as well as the

mean and median of the individual trends.

In most cases the results from the individual datasets are highly consistent in particular for the near-global time series. All

datasets indicate significant near-global recovery trends of around half a percent per decade. The trend derived from the NASA

data is a bit lower at $+0.2$ %/decade. The median and mean trends of all datasets agrees here with the trends of the median

timeseries as shown in Figs. 1 and 2. For the narrower zonal bands not all datasets show significant trends after 1996. The NASA

and GSG datasets show low recovery trends in the NH ($+0.3$ and $+0.1$ %/decade, respectively), while all others are between

0.5 and 0.7 %/decade and significant. In the SH all recovery trends agree to within one tenth %/decade ($+0.7$ %/decade), except

for the NOAA dataset showing a somewhat higher trend of $+1$ %/decade.

In the tropics the recovery trends are close to zero with the exception of the GSG and GTO datasets that have very small and

254 barely significant positive recovery trends of $+0.3\pm0.3$ %/decade. The variations in the trend results from the different datasets

is most likely due to some residual drifts in the datasets that are not accounted for in the data merging. With the use of the full

MLR with all terms and with four years added in the timeseries', the ozone trends in the various zonal bands before and after

1996 remain quite similar to the results reported in W18, but uncertainties are slightly reduced.

**5 Latitude dependent total ozone trends**

Latitude dependent trends in steps of $5°$ are shown from $60°$S to $60°$N for all five merged datasets (thin lines) in Fig. 3. The two

thick blue and red lines are the results before and after 1996 from applying the full MLR to the median timeseries including

$2\sigma$ uncertainties shown as error bars. In the extratropics the recovery trends are on the order of $+0.5\%$ with $2\sigma$ uncertainties of

262 about the same magnitude. In the SH the recovery trends continuously increase to nearly $+1.3$ %/decade in the $55°$S -$60°$S band

while in the NH the recovery trends remain unchanged up to the highest latitudes shown. In the tropics recovery trends are

264 close to zero. One notable change from W18 is that the tropical trends during the ODS rising phase are now more negative





**Table 3.** 1979-1996 and 1997-2020 annual mean total ozone trends in various broad zonal bands. Uncertainties are given as $2\sigma$ and trends in bold are have an absolute magnitude equal or larger than $2\sigma$. $r^2$ is the square Pearson correlation between timeseries of observations and MLR and $\chi$ the residual defined as $\chi^2 = \sum_i (\text{obs}_i - \text{mod}_i)^2 / (n - m)$, where $\text{obs}_i$ are the observations and $\text{mod}_i$ the MLR, $n$, the number of data (years) in the timeseries, and $m$, the number of parameters fitted. All results are obtained using the full MLR.

| zonal bands | MLR/ (2017-2020) minus (1964 -1980) | | median | NASA | NOAA | GSG | GTO | WOUDC |
|---|---|---|---|---|---|---|---|---|
| 60°S-60°N near global | full −2.3% | trend ≥1996 [%/dec.] | **0.4(2)** | **+0.2(2)** | **+0.5(3)** | **+0.4(3)** | **+0.5(3)** | **+0.6(3)** |
| | | trend <1996 [%/dec.] | **−1.5(6)** | **−1.2(7)** | **−1.5(7)** | — | — | **−1.1(7)** |
| | | $r^2$ | 0.94 | 0.94 | 0.93 | 0.92 | 0.93 | 0.89 |
| | | $\chi$ [DU] | 1.1 | 1.1 | 1.2 | 1.3 | 1.2 | 1.3 |
| | | mean trend >1996 [%/dec.] | | | | **+0.4(3)** | | |
| | | median trend >1996 [%/dec.] | | | | **+0.4(3)** | | |
| 35°N-60°N NH | full −3.6% | trend ≥1996 [%/dec.] | **+0.5(5)** | +0.3(5) | **+0.7(5)** | +0.1(6) | **+0.6(6)** | **+0.6(6)** |
| | | trend <1996 [%/dec.] | **−1.9(13)** | **−1.5(12)** | **−1.9(12)** | — | — | **−1.9(15)** |
| | | $r^2$ | 0.88 | 0.90 | 0.89 | 0.88 | 0.87 | 0.85 |
| | | $\chi$ [DU] | 2.9 | 2.7 | 2.7 | 3.0 | 3.0 | 3.3 |
| | | mean trend >1996 [%/dec.] | | | | +0.5(6) | | |
| | | median trend >1996 [%/dec.] | | | | **+0.6(6)** | | |
| 20°S-20°S tropics | full −1.1% | trend ≥1996 [%/dec.] | +0.2(3) | −0.2(3) | +0.1(3) | **+0.3(3)** | **+0.3(3)** | +0.4(5) |
| | | trend <1996 [%/dec.] | **−1.1(7)** | **−1.2(7)** | **−1.0(7)** | — | — | −0.6(12) |
| | | $r^2$ | 0.86 | 0.89 | 0.87 | 0.85 | 0.82 | 0.7 |
| | | $\chi$ [DU] | 1.2 | 1.1 | 1.1 | 1.2 | 1.3 | 1.9 |
| | | mean trend >1996 [%/dec.] | | | | +0.2(3) | | |
| | | median trend >1996 [%/dec.] | | | | **+0.3(3)** | | |
| 35°S-60°S SH | full −4.7% | trend ≥1996 [%/dec.] | **+0.7(6)** | **+0.6(6)** | **+1.0(7)** | **+0.7(7)** | **+0.8(6)** | **+0.8(7)** |
| | | trend <1996 [%/dec.] | **−2.5(16)** | **−2.5(16)** | **−2.4(17)** | — | — | **−2.6(19)** |
| | | $r^2$ | 0.87 | 0.88 | 0.89 | 0.87 | 0.88 | 0.82 |
| | | $\chi$ [DU] | 3.0 | 3.0 | 3.1 | 3.1 | 3.0 | 3.6 |
| | | mean trend >1996 [%/dec.] | | | | **+0.8(7)** | | |
| | | median trend >1996 [%/dec.] | | | | **+0.8(7)** | | |

bold numbers: statistical significance at $2\sigma$

.

(down to $-1\,\%$/decade) while before they were mainly close to zero. This may be caused by the additional proxy terms used

in this study.

After 1996 all trends of all datasets are in good agreement to within $\pm0.3\,\%$/decade. There are some notable differences in

the northern subtropical and northern tropical trends for the WOUDC data (up to $+1\,\%$/decade) compared to the other datasets,

which is most likely caused by larger uncertainties due to the sparsity of ground data at these latitudes. The trend uncertainties

are generally larger for the early period before 1996, which in part may be caused by the different lengths of the periods before

1996 (17 years) and after 1996 (25 years).





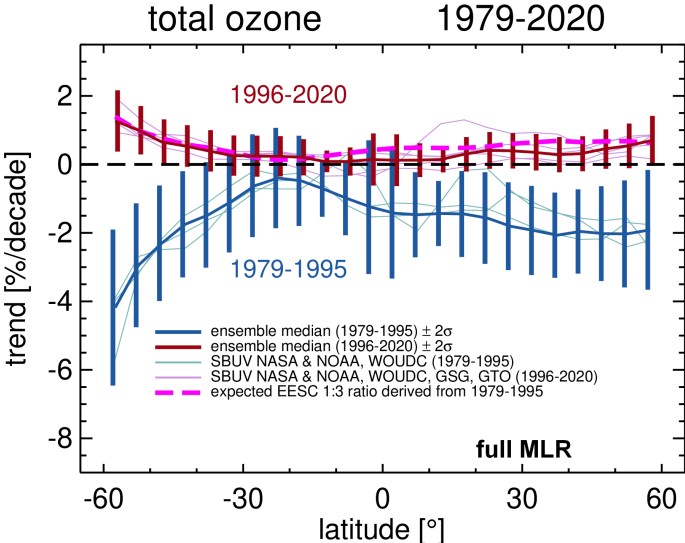

**Figure 3.** Latitude dependent ozone trends in steps of $5°$ from applying the full MLR (Eqs. 1 and 4) to the median timeseries of the five merged total ozone datasets. Trends and $2\sigma$ standard deviations are shown in blue for the time period before 1996 and in red after 1996. The thin lines show the trends of the individual total ozone datasets. The pink dashed lined are the post-ODS peak trends as expected from the 1-to-3 ratio (corresponding to changes in the stratospheric halogen) applied to the median timeseries' trends before 1996.

The dashed pink line shows the expected recovery trends when applying the 1-to-3 ratio (corresponding to the rate change of the EESC) to the trends before 1996. It agrees quite well in the extratropics with the independent linear trend estimates and therefore give us confidence that ozone is responding to the long-term ODS decline. The expected tropical recovery trends are slightly positive while the MLR regressions suggest rather near zero trends, but they still agree within their uncertainties.

In order to elucidate further on the interpretation of the independent linear trends after 1996 as recovery trends, we repeated the analyses using the standard MLR which excludes several terms responsible for changes in atmospheric dynamics and transport (Eq. 1 with $P(t) = 0$). The latitude dependent trends from the standard MLR are shown in Fig. 4. While the recovery trends are nearly unchanged in the SH, the NH recovery trends are reduced to zero in the NH extratropics. On the other hand the tropical trends before 1996 are closer to zero. The expected recovery trends (from the 1-to-3 EESC ratio) have become larger with increases to $+1.5\,\%$/decade at the higher latitudes now in both hemispheres. The most obvious result is that the independent linear trends after 1996 in the NH being close to zero now clearly deviate from the expected 1-to-3 ratio. It appears that the additional atmospheric dynamics terms in the regression balance the positive recovery trends from the full MLR which explains why total ozone in the NH appears more or less stable during the last two decades (panel a of Fig. 2).

The declining trends in the NH before 1996 (Fig. 4a) are stronger in the standard MLR and are comparable to the SH (about $-4\,\%$/decade near $60°$ latitude). On the other hand ODS related trends are expected to be somewhat stronger in the SH as the influence from polar ozone losses on mid-latitude ozone is thought to be larger in the SH, since Arctic ozone losses are more sporadic and generally smaller. In that regard the trends from the full MLR seem to support this notion.





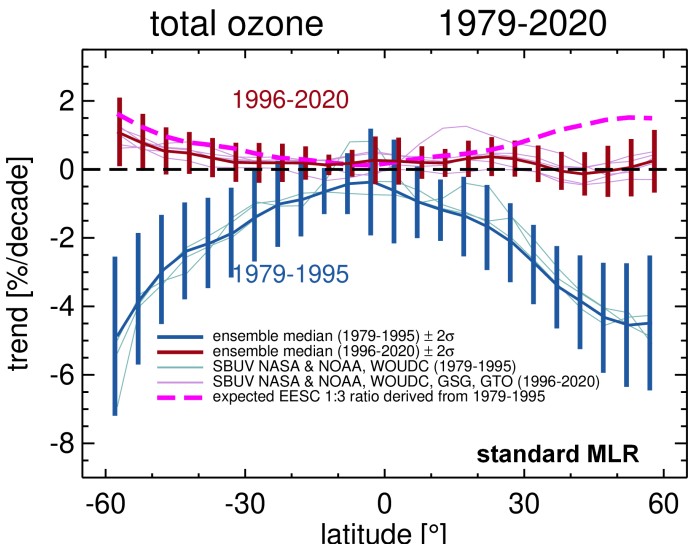

**Figure 4.** Same as Fig. 3, but from applying the standard MLR (Eq. 1 and $P(t) = 0$)).

The comparisons of trends from the standard and full MLR reveal that the NH ozone recovery is balanced by long-term changes in atmospheric dynamics (circulation and transport changes) or in other words the near zero linear post ODS-peak trends are caused by the combination of ODS-related recovery and dynamical changes. These two signals are more clearly separated in the full MLR. Before discussing this further, we will take a look at the contributing factors or terms in the MLR. Figure 5 shows the maximum response of the various terms in Eqs. 1 and 4 as a function of latitude (from the fit to the median timeseries).

Well-known factors like solar activity and QBO show the expected behaviour, i.e. more ozone during solar maximum at all latitudes (see e.g. W18) and the opposite sign in the QBO response between inner tropics and extratropics (Bowman, 1989; Baldwin et al., 2001). The solar response is of similar magnitude at all latitudes, which means that the solar effect in the lower stratosphere is mostly indirect via changes in temperature and associated atmospheric circulation changes (e.g. Dhomse et al., 2021).

In the NH the BDC and and AO mostly contribute to ozone variability. Interestingly, there is an influence from the BDC from one hemisphere to the other in both directions. BDC-N results in opposite responses in the tropics and NH extratropics. This is expected from the planetary waves driving the BDC leading to ascent in the tropics (lower ozone) and descent in the polar region (higher ozone) (e.g. Randel et al., 2002; Weber et al., 2011). The correlation of ozone anomalies in the NH winter/spring to SH total ozone was reported by Fioletov and Shepherd (2003) and is believed to explain the positive response in SH total ozone. Somewhat surprising is the impact of the SH BDC on NH ozone with a negative ozone response, for which we have no explanation.

The major volcanic eruption of Mt. Pinatubo in 1991 had a stronger impact on the NH reducing ozone for several years after the event, while ozone advection apparently balanced the surface acid particle (aerosol) related ozone losses in the SH (Schnadt

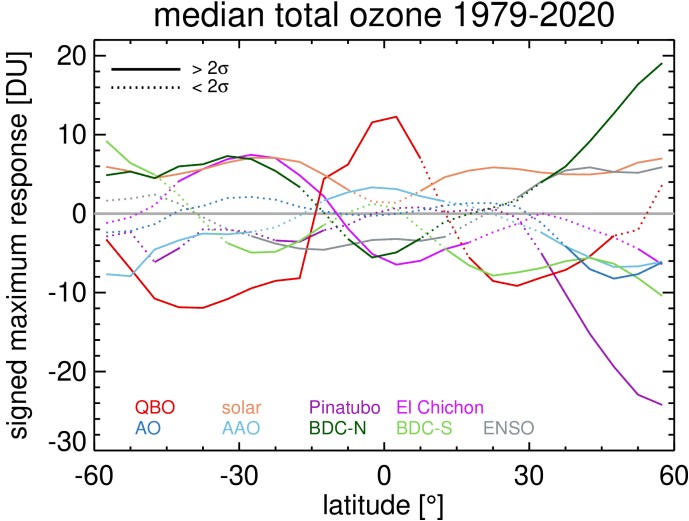

**Figure 5.** Signed maximum timeseries contribution from various terms in the full MLR equation applied to the median of the five merged total ozone datasets. Solid line indicates values of fit coefficients that are larger than their $2\sigma$ uncertainty. The sign of the BDCs proxy time series is reversed, so that in both hemispheres positive BDC term values correspond to enhanced Brewer-Dobson circulation. Negative values an anti-correlation of the ozone response to the proxy. For instance, positive solar contributions mean high solar activity leads to more ozone.

Poberaj et al., 2011; Aquila et al., 2013; Dhomse et al., 2015). The second large major volcanic eruption from spring 1982 lead to aerosol related ozone loss in the tropics and NH, while surprisingly a positive ozone response in the SH is seen possibly related to some atmospheric circulation changes compensating chemical effect from the El Chichon eruption. In contrast to Mt. Pinatubo, which spread sulfuric acid particles into both hemispheres, enhanced aerosols from El Chichon were confined to lower latitudes in the NH (McCormick and Swissler, 1983) consistent with the region of negative ozone response shown in Fig. 5.

The main reason for stable ozone levels observed in the NH since 2000 were identified to stem from the balancing of the positive observed recovery trend by negative trends due to circulation changes and ozone transport (see Figs. 2 and 3) The change in the BDC-n proxy and AO over the last 55 years is shown in Figure 6 along with March total ozone northward of 40°N . The variability in the extratropical annual mean is usually dominated by the variability in winter/spring, where BDC maximizes in the seasonal cycle. Apart from the strong drop in ozone in the 1990s related to the major volcanic eruption and associated circulation changes, NH total ozone has been steadily declining over the last 55 years (about 25 DU). This decline is coherent with an overall positive shift of the AO index. A weakening of the BDC is also seen but appears less clear than for the AO.

A positive shift in the AO and a weakening of the BDC results in a strengthening of the polar vortex, which is associated with larger polar ozone losses (Lawrence et al., 2020). Hu et al. (2018) linked a recent strengthening of the stratospheric Arctic vortex in part to a warming of sea surface temperatures in the central northern Pacific. A recent downward trend in lower stratospheric ozone has been reported by Ball et al. (2018) that could be consistent with the total ozone observations. Other

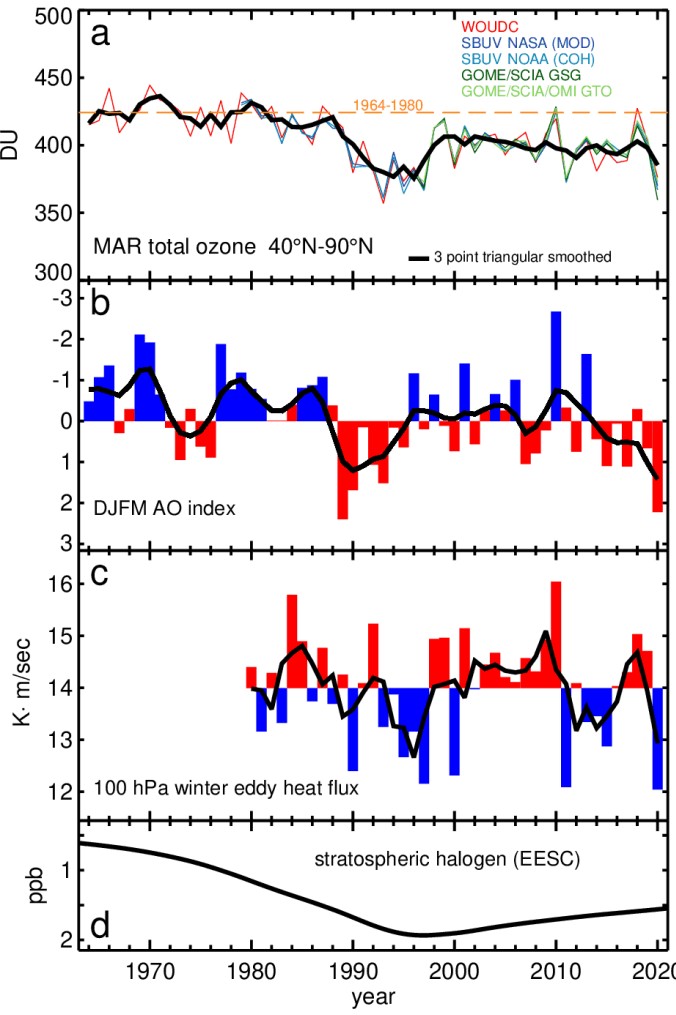

**Figure 6.** Panel a: March NH total ozone (40°N -90°N ) from the five bias-corrected merged datasets (colored) and the smoothed median timeseries (thick black line). Panel b: DJFM Arctic oscillation (AO) index. Black lines shows the three-point triangular smoothed timeseries. Note the inverted y-scale series. Panel c: 100 hPa winter eddy heat flux September to March average (BDCn proxy) with black line showing the three-point triangular smoothed timeseries. Panel d: Inverted stratospheric halogen timeseries in ppb representative for middle latitudes (Newman et al., 2007).

studies with many different ozone profile datasets did not show significant trends in the lower stratosphere due to very large
variability and lower accuracy of the satellite data in this altitude region (Sofieva et al., 2017; Steinbrecht et al., 2017; Arosio et al., 2019).





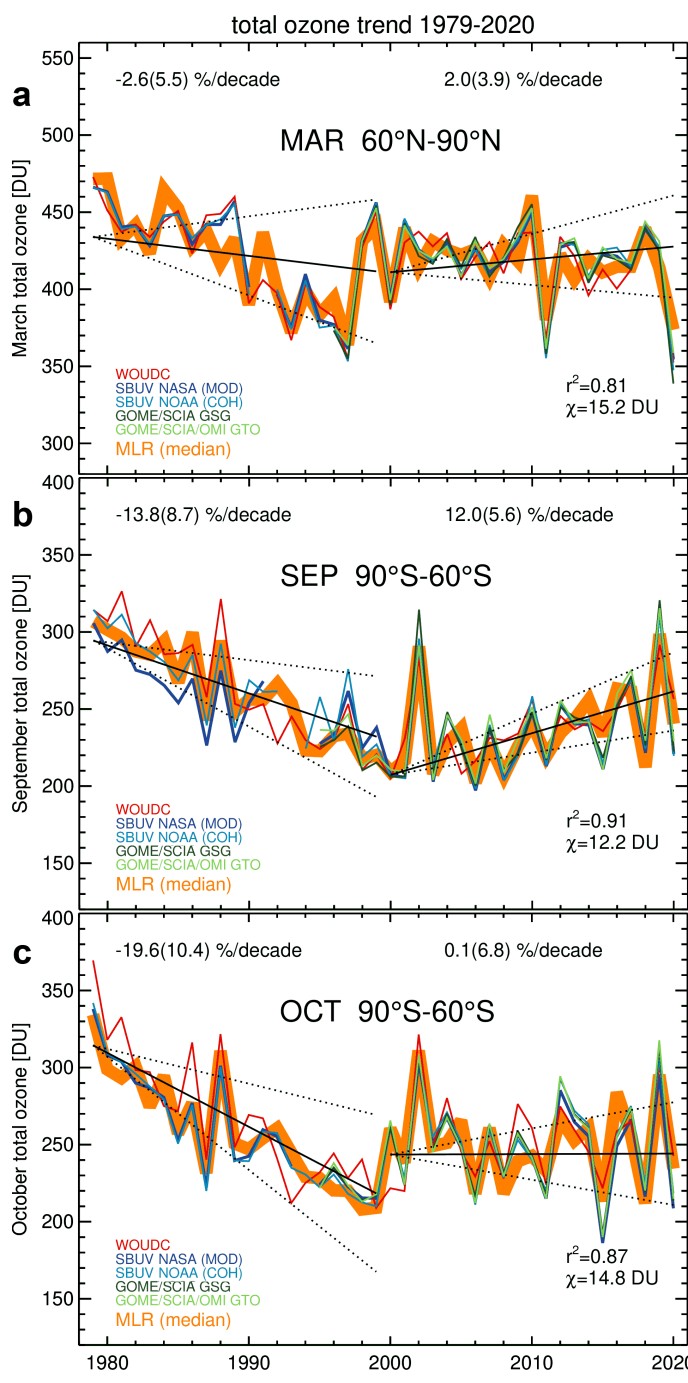

**Figure 7.** Polar ozone trends derived from the full MLR applied to the median timeseries. a) March 60°N-90°N, b) September and c) October 60°S-90°S. see Fig. 1 for more details.


## 6 Trends in polar spring

Earlier signs of ozone recovery has been observed in September above Antarctica (Solomon et al., 2016, W18). Now with four more years of data this recovery of about 11 %/decade remains robust (see panel b of Fig. 7). During September the Antarctic ozone hole size usually increases and reaches its maximum in late September and early October. In a typical Antarctic winter, ozone is completely destroyed in the lower stratosphere, which may explain why no recovery is yet observed in October over the polar cap (panel c in Fig. 7). Despite the lack of recovery in the late ozone hole season, several diagnostics clearly indicate a healing of Antarctic ozone as a consequence of the Montreal Protocol. Stone et al. (2021) show that the onset of the Antarctic ozone hole has been shifted to later dates despite the larger than average ozone holes observed in recent years (e.g. 2015 and 2020).

Panel a of Fig. 7 shows the March ozone timeseries above the Arctic with ozone recovery trends not statistically different from zero. The trend results in the polar regions as shown in Fig. 7 basically confirm the results from W18 and Langematz et al. (2018). Table 4 summarises the polar trends for the individual datasets and the mean timeseries. Within the trend uncertainties all datasets are in very good agreement.

## 7 Summary and conclusions

We derived globally total ozone recovery trends from five merged total ozone datasets using a multiple linear regression with independent linear trend (ILT) terms before and after the turnaround in stratospheric halogens in the middle 1990s (∼2000 in the polar regions). When properly accounting for dynamical changes via atmospheric circulation and transport, these retrieved trends may be interpreted as recovery trends related to changes in the stratospheric halogens as a response to the Montreal Protocol and Amendments phasing out ozone depleting substances.

For the near-global average we see small recovery trends of about 0.5 %/decade with main contributions from the extratropics in both hemispheres. The ratio of ozone trends after and before the turnaround year is in very good agreement with the trend ratios in stratospheric halogens.

In the tropics recovery trends are not statistically different from zero. In line with earlier observations (Solomon et al., 2016, W18), polar ozone recovery has been only identified in September above Antarctica, which is connected to the observed delay in the onset of the Antarctic ozone hole (Stone et al., 2021). In the Arctic large interannual variability still prevents the detection of early signs of recovery.

Although we showed that ozone recovery is evident at NH middle latitudes the total ozone levels in the NH extratropics have been more or less stable since about 2000. Our regression results show that the recovery here is balanced by changes in ozone transport. A long-term positive drift in the AO index over the last 55 years is indicative of a strengthening of the Arctic vortex (Hu et al., 2018; Lawrence et al., 2020; von der Gathen et al., 2021) and reduced winter/spring transport of ozone into middle and high latitudes. This result may be consistent with the observed decline in lower stratospheric ozone in the extratropics as reported by Ball et al. (2018) and Wargan et al. (2018). Other studies and datasets, however, do so far not confirm the long-term decline in the lower stratosphere Arosio et al. (2019); Steinbrecht et al. (2017); Sofieva et al. (2017), which may be in part due





**Table 4.** Polar total ozone trends in March (NH), September (SH), and October (SH) before and after 2000. Uncertainties are provided for $2\sigma$ and trends in bold indicate statistical significance. $r^2$ is the square Pearson correlation and $\chi$ the residual (see caption of Table 3). The results were obtained from the full MLR.

| zonal bands | MLR | | median | NASA | NOAA | GSG | GTO | WOUDC |
|---|---|---|---|---|---|---|---|---|
| 60°N-90°N | full | trend $\geq$2000 [%/dec.] | +2.0(39) | +3.0(35) | +3.4(37) | +3.1(40) | +3.6(37) | +1.3(44) |
| March | | trend $<$2000 [%/dec.] | −2.6(55) | −0.3(51) | 0.0(54) | — | — | −2.3(61) |
| | | $r^2$ | 0.81 | 0.85 | 0.85 | 0.84 | 0.84 | 0.76 |
| | | $\chi$ [DU] | 15.2 | 13.2 | 14.0 | 14.9 | 13.6 | 17.0 |
| 60S°S-90°S | full | trend $\geq$2000 [%/dec.] | **+12.0(56)** | **+11.0(65)** | **+10.1(68)** | **+12.2(57)** | **+11.2(57)** | **+10.9(62)** |
| September | | trend $<$2000 [%/dec.] | **−13.8(87)** | **−8.9(100)** | **−11.6(105)** | — | — | **−19.1(107)** |
| | | $r^2$ | 0,91 | 0.85 | 0.87 | 0.92 | 0.91 | 0.89 |
| | | $\chi$ [DU] | 12.2 | 14.2 | 14.6 | 12.2 | 12.1 | 13.3 |
| 60°S-90°S | full | trend $\geq$2000 [%/dec.] | +0.1(68) | −1.9(68) | +0.1(65) | +0.9(71) | +0.5(72) | +4.1(91) |
| October | | trend $<$2000 [%/dec.] | **−19.6(104)** | **−19.4(104)** | **−21.0(100)** | — | — | **−18.9(138)** |
| | | $r^2$ | 0.87 | 0.87 | 0.88 | 0.86 | 0.86 | 0.81 |
| | | $\chi$ [DU] | 14.8 | 14.8 | 14.2 | 15.5 | 15.7 | 19.7 |

bold numbers: statistical significance at $2\sigma$

.

to the larger uncertainties of satellite observations in this altitude region. From chemistry climate-models it is expected that
the BDC and ozone will increase as a result of greenhouse gases and models so far cannot explain the extratropical decline in
lower stratospheric ozone (Dietmüller et al., 2021).

Another point which will be important is to show consistency between total ozone trends and both stratospheric and tropospheric ozone trends. The ozone satellite datasets still show significant differences and opposite signs in trends (Gaudel et al., 2018).

*Data availability.* The sources of the various datasets and proxy time series (explanatory variables) used in this study are summarised in Tables 1 and 2.

*Competing interests.* No competing interests are present.

*Acknowledgements.* M.C.E., D.L., and C.A. are grateful for the support by the ESA Climate Change Initiative project CCI+. M.W. and J.P.B. acknowledge the financial support of the State of Bremen. S.M.F. is supported by the NASA Long Term Measurement of Ozone program WBS 479717.



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
