# Peer review of "Global total ozone recovery trends attributed to ODS changes derived from five merged ozone datasets"

_Atmospheric Chemistry and Physics, 2021_

## Author Comment (AC1)

**Reply to Reviewer #2**

Weber et al., Global total ozone recovery trends attributed to ODS changes derived from five merged ozone datasets, doi:10.5194/acp-2021-1058

Reviewer comments are provided here with our replies written in italics.

The manuscript "Global total ozone recovery trends derived from five merged ozone datasets" by M. Weber provides an update to a study published by the first author in 2018, with four more years of data added to the five analyzed datasets (four satellite datasets and one dataset comprised of ground-based measurements). A multiple linear regression is applied to annual mean data from the period 1979 to 2020 to determine total column ozone (TCO) trends in different broad latitudinal bands for the period in which concentrations of ozone-depleting substances (ODSs) increased in the atmosphere, and for the period after the peak concentrations had been reached. The multiple linear regression includes next to the typical proxies also several dynamical variables (e.g. a proxy for the Brewer-Dobson circulation (BDC) or the Antarctic/Arctic Oscillation (AAO/AO)) which is one of the main differences to other trend analyses based on TCO data. The authors find with this method significant positive trends (related to the reduction in ODSs in the atmosphere) for the period 1997-2020 for the near-global mean (60S-60N), as well as for the Northern hemisphere mid-latitudes for which the trend is near zero if the dynamical proxies are not included in the regression.

The manuscript is very well written and well structured, mostly the data and methods are explained in enough detail to allow the reader to understand what is going on (in a few cases I found the description slightly too short and I have mentioned them in the details below), and the topic lays clearly within the scope of the ACP journal. There are a few minor things that I commented about below that are easy to fix, but there are two main points that I think need careful adjustment of the manuscript or some additional thought.

→ *We address these points (see specific replies below).*

I recommend the publication of the manuscript after revisions.

Two main points:

Attempting an attribution with a multiple linear regression that includes non-orthogonal proxies is tricky. Especially if several proxies include a trend. The hope then is, that the regression is able to separate the trend contribution from the different proxies based on the additional variability the proxies provide. However, it is possible that trends are not assigned correctly to the different proxies which would falsify the signal of the trend that if of interest, in this case here, the trend caused by ODSs and not by changes in dynamical variables. The authors argue that with the addition of the dynamical proxies the variability of the time series' are matched better by the regression results. There are two points that make me somewhat doubtful of this statement: (1) the pre-1996 trends change clearly with the introduction of the dynamical proxies (Figures 3 and 4) although the main trend signal should be coming from ODS-related changes in this period; (2) the signal from the SH Brewer Dobson circulation proxy in the NH polar regions that cannot really be explained. I think the manuscript needs more discussion of these points to strengthen the claim that the addition of the dynamical proxies can indeed robustly
isolate the ODS-related trends. For the first point I raised I would suggest to check the older literature
      about regression results for the pre-1996 period where dynamical proxies have been used. I have added
two references in the comments below that might be worth checking out. And there might even be
      more that could be checked and where the results could be compared to the pre-1996 ODS-related
trends calculated here. For the second point I raised I think it would be helpful to do some sensitivity
      test to check the robustness of the trend results and the contribution of the individual proxies: (I) not
using the trend proxy but JUST the dynamical proxies, how do their contributions change if at all; (II) use
      some of the dynamical proxies only in the regions where they occur, e.g. AAO only in the SH, AO only in
the NH, etc.; how does the contribution of these proxies change (if at all), and how does the ODS-related
      trend change? I think these sensitivity test will go a long way to show the robustness of the results
presented here in this manuscript.

      → *We added two new tables to summarise the results from new sensitivity tests we carried out. New*
*Table 4 shows different MLR settings applied to the median total ozone timeseries in broad zonal bands*
      *(as defined in Table 3). Here the results from the standard and full MLR are listed. In addition, we applied*
*an iterative MLR approach where statistically insignificant terms (2sigma criterion) are successively*
      *excluded before the final MLR run. In order to document the changes from the MLR fits to the period up*
*to and including 2016 as in W18, the results of the different MLR settings applied to the current data for*
      *the shorter period is provided in Table S1 (Supplement). Note that the results in Table S1 may differ from*
*W18 as the merged datasets have been updated and data before 2017 may have changed as well.*

      *The following can be concluded from these additional sensitivity tests:*

*"The inclusion of the dynamical proxies generally improved the MLR fit (r2 and chi values). Except for the*
      *NH zonal band (35N-60N) the various MLR settings yield nearly the same post ODS-peak trends for all*
*broad zonal bands (new Table 4). There are, however, larger changes in the trends before the middle*
      *1990s. In the extratropics the early-period trends are lower (-4.0%/decade vs. -1.9%/decade in the NH*
*and -3.1%/decade vs. 1.9%/decade in the SH) in the standard retrieval. This means that atmospheric*
      *dynamics and transport changes contributed to lower early-period extratropical total ozone trends in the*
*standard regression (due to the lack of these dynamical terms in the MLR). The opposite is the case in the*
      *tropics where the early-period trends in the standard MLR are slightly higher than in the full MLR. This*
*opposite behavior is consistent with ozone transport patterns due to the Brewer-Dobson circulation.*

      *The only significant changes in the post ODS-peak trends are seen in the NH extratropics. In the standard*
*MLR this trend is zero, while the full and iterative MLR show trends of a half per cent per decade. The*
      *sum of the ODS-related trend (full MLR) and atmospheric dynamics contribution (difference in the trends*
*between full and standard MLR) cancel to result in a zero trend in the standard MLR. The negative*
      *dynamical trend contribution in the NH is further discussed later in the paper. The correlation between*
*regression and observations are substantially lower in the standard retrieval (r2=0.74 vs. 0.88) which*
      *means that the standard MLR seems not to capture all variability and changes in total ozone.*

*The results shown in Table 4 are compared with the results from the MLR applied to the period limited up*
      *to 2016 (same period as in W18) as shown in Table S1 (Supplement). Results from the shorter time period*
*are nearly identical to those shown in Table 3. There is one notable change. The uncertainties of the NH*
      *trends from the full MLR up to 2020 are reduced such that these trends have become barely significant*

*(2sigma). The Post-ODS-peak trend of the standard MLR is slightly positive up to 2016 but statistically insignificant and within the uncertainties not different from the current results."*

I think it is really important to clarify throughout the manuscript (including the title!) what kind of trends
the authors talk about. Mostly, the trends that are discussed are the trends that are attributed to the reduction of ODSs in the atmosphere WITHOUT any contribution of dynamics to the trend. In many
places this is not totally clear since the trends are only called "recovery trends". However, for me this is the main point of the manuscript and the difference to other studies. It would therefore be extremely
important and very helpful if the authors could be more specific in how they name the trends throughout the manuscript (e.g. instead of referring in the abstract in line 11 to "The near global trend
of the median of all datasets…" it would be better to be more specific and refer to "The near global ODS-related trends …", and specifying this in the title like "Global total ozone recovery trends attributed to
ODS changes derived from five merged ozone datasets")

→ *We agree. The title has been changed accordingly and we made appropriate changes in the text in*
*order to refer to ODS-related rather than recovery trends.*

Minor comments:

Line 10: "… is indeed on slowly recovering…" – remove the "on". → *done*

Line 16: data from which phase of CCMI? Please specify. → *add "(Phase 1 CCMI REF-C2 scenario)"*

Line 71: It is not clear in this section what the spatial coverage of the described datasets is. I assume 90S-90N since also polar regions are analyzed. Please add this information to the dataset descriptions.
→ *added at the end of the paragraph (l. 82): "All datasets cover the entire earth except for months and latitudes under polar night conditions (winter months)."*

Line 72: "ground-based" instead of "ground" → *done*

Line 78: "ground based Brewers, …" - remove the "ground based" since it is already mentioned at the
beginning of the sentence. → *done*

Line 80: Add also here the information from which phase of the CCMI project simulations was analyzed.
→ *changed to "Phase 1 CCMI Initiative" (add "Phase 1")*

Line 129: It is not clear how and by whom the ground-based dataset was updated. The references for
the dataset are relatively old, therefore it would be good to add a few words on how the dataset was updated to the year 2020. → *The data set is a data product provided by the WOUDC and updated*
*regularly. It is available from https://woudc.org/archive/Projects-Campaigns/ZonalMeans/. We added this information after the text in line 131: "The data set is a data product provided by the WOUDC and*
*updated regularly"*

Line 135: The word "belt" is used here, although it is only explained in the following sentence what
exactly is meant by it. This should be switched to make it clearer for the reader what is meant by "belt".
→ *We replaced the corresponding sentences (lines 134-137) with "Then, for each station and for each*
*month the deviations from the climatology were calculated, and a zonal mean value for a particular month was estimated as a mean of these deviations. The calculations were done for 5◦-wide latitudinal*

*zones. In order to take into account various densities of the network across regions, the deviations of the*
*stations were first averaged over 5° by 30° cells, and then the zonal mean was calculated by averaging*
*these first set of averages over the 5°-wide latitudinal zone."*

Line 154: the data were bias-corrected. It would be nice to give here a range of biases that needed to be
adjusted. I understand that the biases can be different for the broad latitude bands and datasets, but
some kind of number/range would be nice here. → *The various biases between datasets are irrelevant*
*and do not change the derived trends.*

Line 169: "applies" should be "apply" → *done*

Line 175: "." is missing after the parenthesis. → *done*

Line 176: The year 1996 is the time for maximum EESC concentrations for which region of the globe?
Tropics? Everything besides the polar regions? → *"... and some years later (t0=2000) in the polar*
*regions" is replaced by "except for the polar regions (>60°) where t0=2000" and removed the next*
*sentence.*

Line 177: It would be good to give the exact latitude ranges here which define the polar regions. → *see*
*the previous comment.*

Line 190: The end of the sentence is slightly misleading. I would add "for these years" before "were
calculated" to clarify that only for the years with too many missing data no annual means were
calculated. → *change second sub phrase after "and" to: "and for these years annual mean data were*
*treated as missing data,"*

Line 226: What about the pre-1996 trends? Did they stay very similar to W18 as well? → *see reply to the*
*general comment above.*

Line 248: "agree" instead of "agrees" → *done*

Line 255-257: It might be nice to add here a table with the trends reported from W18 and calculated
here. It would provide a nice overview of things that changed and things that stayed roughly the same
(just for the multi-observational median, not each individual dataset) → *see reply to the general*
*comment above and New Table 4 and S1.*

Line 269: "ground-based" instead of "ground"? → *done*

Line 285: Are there any studies that report on trends pre-1996 based on regression methods that use
also dynamical proxies? There is one looking at ozone soundings at Payerne (Weiss et al., JGR, Vol. 106,
D19, 22685-22694, 2001), and one looking at individual TCO station measurements (Maeder et al., 2007,
https://doi.org/10.1029/2006JD007694) but there might be even more analyzing total column ozone
data with dynamical proxies. As mentioned above, I think it would be helpful to provide an estimate
how well the ODS-related trends compare with earlier findings for the pre-1996 period since they did
change quite a bit with the introduction of the dynamical proxies. → *see reply to the general comment*
*above. The older studies mainly used a piecewise linear trend (PLT) model and thus are difficult to*
*compare. In W18 we discuss the various trend models and our decision to use preferably the ILT method*
*in W18 (and this study).*

Line 305/306: Couldn't this signal be a spurious regression result where the attribution did not work properly between the trend proxy and the dynamical proxies also including a trend? I think some sensitivity test (as mentioned above) would be helpful here to test the robustness of this signal. → MARK (see general comments) → *see reply to the general comment above. It appears that the post-ODS trends are in most cases unchanged regardless of the number of extra terms used in the MLR. The linear trend term is the only low-frequency term in the MLR equations, while the dynamical proxies have some high-frequency contributions. This makes the trend estimates rather robust and less sensitive to the various other terms used in the MLR.*

Line 316: "." missing after the parenthesis. → *done*

Line 331: "have" instead of "has" → *done*

Line 366-368. This sentence seems somehow out of place here. I think it needs a little more explanation and detail. → *We omit this sentence, as we did not discuss the possible impact of tropospheric ozone on column trends. The impact is possibly rather small when using annual and zonal means.*

---

## Author Comment (AC2)

**Reply to Reviewer #1**

**Weber et al., Global total ozone recovery trends attributed to ODS changes derived from five merged ozone datasets, doi:10.5194/acp-2021-1058**

Reviewer comments are provided here with our replies written in italics

1. Short resume

Weber et al. present a comprehensive analysis of trends in total ozone, focusing primarily on the period
since the turnaround in ozone-depleting substances. This is an update and extension of earlier work
published in 2018. In contrast to latter publication, the authors now claim the detection of increases
(0.4%/decade) in near-global (60S--60N) total ozone since 1996, with high confidence ($>3\text{-}4\sigma$).
Positive trends over broad mid-latitude region in both hemispheres (35N--60N and 35S--60S), about 0.5-
-0.7%/decade, are significant as well although close to the 2sigma detection threshold.

The dynamical process terms (Arctic and Antarctic Oscillation, Brewer-Dobson circulation) in the
regression model play a central role in this detection, especially at northern mid-latitudes. The authors
deliberately chose not to detrend the dynamical terms prior to regression, in order to account for any
long-term changes in AO, AAO and BDC. In doing so, they find that trends become less negative before
and more positive since 1996 across large regions of the low- and mid-latitudes. This more
complete attribution results in a higher significance of the trends, especially in the northern hemisphere
where the 2sigma detection threshold was passed. Hence, the authors conclude that dynamical changes
appear to counterbalance the recovery of ozone in the mid-latitude NH.

The authors furthermore explain the positive recovery trend of total ozone as a result of changes in
ozone-depleting substances. Indeed, the ratio of the rate of increase and decrease in ODS
concentrations is consistent with the rate of depletion and recovery of total ozone across all 5° latitude
bands between 60S and 60N.

2. Recommendation

This paper provides an important update to previous assessments of long-term changes in total ozone. It
is very well written and accessible to a large scientific audience. The methodology is sound and the
presented results support the claims made by the authors. I highly recommend publication of this work
in ACP if my remarks below have been addressed.

3. Major comments

Ordered in order of appearance in the text.

3.1 Extension of GOME-type backwards in time (Sect. 2.7)

I understand the importance of covering a sufficiently long period, but is this backwards extension for
GOME-type data records still needed now that more than two solar cycles have been completed since
1995? Doesn't this break the independence between SBUV and GOME-type estimates? By how much
does the negative trend in the SBUV period influence the recovery trend estimates during the GOME-
type period? Have you tested the sensitivity of the resulting trend to the choice of NASA COH or NASA
MOD, and without the extension?

→ *We try to stay consistent with the W18 paper, where we also applied for these extensions. The main*
*idea of extending to the full-time period is to have as close as possible the same impact from all proxy*
*terms, not only the solar term, during the full-time period. Since we use independent linear trends before*
*and after the ODS peak, the impact of early trends on the late period trends is minimised.*

Avoiding data gaps is important but preserving data quality/stability is perhaps even more important
under high aerosol backgrounds. Could you elaborate why gaps are more important or, if that is not the
case, comment on the stability of both SBUV records after Pinatubo?

→ *Calculated annual means were accepted as valid if at least 80% of the monthly means were*
*contributing (10 months minimum) and 80% of the 5° zonal means were available in the broader zonal*
*bands. If these conditions are not met we consider the annual mean as not representative for the given*
*year and should be excluded from the MLR. The consistency of the SBUV data records with other total*
*ozone data has been documented, e.g. Chou et al, 2014.*

3.2 No reference to how trend errors are estimated (Sect. 3)

Many trend estimates (Fig. 3) are close to the 2sigma threshold. The computation of MLR coefficient
uncertainties, therefore, deserves some attention, this is missing right now. Please explain how MLR
parameter errors are computed or refer to relevant publications. → *All uncertainties are given as 2sigma*
*and sigma is calculated from the least-squares fit. This is a standard approach and is described in many*
*statistics textbooks.*

Somewhat related to this, was there any consideration of including reported measurement errors in the
regression? → *Measurement errors were not accounted for. Not for all merged datasets, an uncertainty*
*estimate is provided from merging the data.*

3.3 Annual time series p.7, l.180: Could you motivate the choice for analysis of annual mean time series instead of monthly
mean data? Is there an impact on the trend estimates and their significance? Please refer to relevant
publications. → *The main reason for using annual means is that this does not require corrections for*
*auto-correlation (mentioned in the text). Adding auto-correlation terms in the regression will not alter*
*the trends but increases uncertainties in the fit coefficients (and trends). The short-term variability is not*
*the focus of our MLR except that we try to minimise the residual of the regressed timeseries.*

3.4 Robustness of attribution to dynamical processes (Sect. 5)

Previous work by the authors (Weber et al, 2018) also considered terms for dynamical processes in the MLR. At the time, however, no significant positive trends were detected (Fig. 9). → *see our general comment in the beginning of the reply to reviewer #2*

It would be enlightening to discuss whether the four additional years of data have truly helped to attribute ozone changes more robustly to dynamical changes. Or, whether it is plausible that the current attribution is subject to geophysical variability (and measurement uncertainty). → *see our general comment in the beginning of the reply to reviewer #2*

4. Minor comments p.1, l.12-13: Near-global trend values disagree with quoted values in Section 4. Please revise. →
*Numbers in the abstract have been adjusted to the values shown in Fig. 1.*

p.3, l.82: "Annual mean timeseries of all five merged datasets are in very good agreement". Somewhat
subjective, please add a number. → *add: "… to within a few DU"*

p.5, l.132: The evolution in satellite quality has been described adequately. This is missing in the WOUDC
section. Surely, there must have been progress in the calibration of these instruments or the coherence of the network since the work by Fioletov in 2008. If so, could you update this section accordingly? →
*The ground-based network calibration procedures have been established a long time ago and there are no major changes in the network operation. The same is true for the WOUDC data set that is regularly*
*generated by the WOUDC. We added a reference to a recent paper where the differences between satellite and ground-based data are discussed on a global scale. We added a reference to Garane et al.*

p.6, l.142-143: "[...] can be estimated with a precision comparable with satellite-based data sets (~1%)."
A reference would be appropriate. → *Comparison of satellite and ground-based data sets is discussed in the following paper: Chiou, E. W., Bhartia, P. K., McPeters, R. D., Loyola, D. G., Coldewey-Egbers, M.,*
*Fioletov, V. E., Van Roozendael, M., Spurr, R., Lerot, C., and Frith, S. M.: Comparison of profile total ozone from SBUV (v8.6) with GOME-type and ground-based total ozone for a 16-year period (1996 to 2011),*
*Atmos. Meas. Tech., 7, 1681–1692, https://doi.org/10.5194/amt-7-1681-2014, 2014. It was referenced in line 140 and we added this reference to line 143 (at the end of the last sentence)*

p.6, l.150: Remove "from the past into the future" as the statement "between 1960 and 2100" is more
than sufficient. → *done*

p.6, l.154-156: I am sorry, I did not get the point of "The multi-dataset mean was then added back to
each dataset, such that all bias corrected timeseries are provided in units of the total column amounts (W18). However, the trend results derived from them are identical to those derived using anomaly
timeseries." Could this be clarified a bit better for the non-expert? → *This procedure means that the bias-corrected time series differ from anomaly timeseries by a constant offset (multi-instrument mean).*
*The bias correction has no influence on the calculated trends but makes the data more legible in the plots.*

p.6, l.154: "to the mean". The 1998-2008 mean at the global or local level?  → *all data are annual mean zonal means and for each zonal band considered an average for the period 1998-2008 was calculated for*
*each dataset and a mean over all datasets (multi-instrument mean) calculated*

  p.6, l.165: See comment below, the second term in Eq. 1 should be b_1 (t-t_0) → *this is not correct, since*
*t_0-t is positive (t_0>t), b_1 will be negative if ozone declines.*

  p.6, l.166: "coefficients b_1 and [...]" This is inconsistent with the notation in Eq 1. Sign of first trend
term (t0-t) implies that positive b_1 values represent a decline in ozone. Please change this. The factors X_1(t) and X_2(t) define the decline/recovery periods.  → *see previous statement*

p.6, Eq.2 and 3: Figure 1 suggests that the "recovery" period starts in 1996, so the turnaround is defined as $t_0=1996$. If this is correct, then the notation in Eq. 2 and 3 should be changed to $X_1(t)=1$ for $t
< t_0$ and $X_2(t)=1$ if $t \leq t_0$ (and vice versa for $X_i=0$). The trend model is not continuous at $t_0$, hence $<$ or $\leq$ do make a difference.  → *This was indeed not consistent and has been*
*corrected at several places.  The early period is t < t0, late period t >= t0. So the first period includes t_0=1996 and the late period starts with 1996. As mentioned in the text the shift of t_0 back and forward*
*did not change the trend estimates.*

  p.7, l.185-187: Is there any particular reason why you haven't used GloSSAC v2 (Kovilakam et al., 2020)?
→ *We actually tested the Glossac dataset, but we found only negligible differences in the trend estimates. This is likely due to the fact that the El Chichon and Mt Pinatubo eruptions dominate the*
*stratospheric aerosol optical depth proxy timeseries. This effect is even enhanced since we use two proxies to separate both major volcanic events.*

p.8, Table 2: EHF is missing from this list. Where can it be downloaded?  → *As mentioned in the text, the eddy heat flux was calculated by us from the ERA5 reanalysis data and was not taken from an external*
*source. A description of how to derive the eddy heat flux from reanalysis data is given in W18.*

  p.8, Eq.4: "BDCn" and "BDCs" should be explained in the text.  → *added, "The BDCn and BDCs are 100*
*hPa eddy fluxes in the northern (n) and southern hemisphere (s)."*

  p.8, l.208: "the linear trend terms best approximate EESC related trends". Can a match between ozone
trend and EESC expectations really validate the choice of terms in the MLR? There is a risk of a circle reasoning here. If the improved agreement with EESC expectations is motivating the choice of terms in
the MLR model then you can't use this same agreement again to conclude a causal relation between trend and EESC.  → *We only assume that all trends not related to ODS changes are contained in the*
*proxy terms. The linear trend before and after the ODS peak is independent, but It turns out that the trend ratio before and after the ODS peak is consistent with the rate changes of EESC to within the*
*uncertainties from the regression. However, we know that there are feedbacks between ODS (ozone) and climate (dynamics). Therefore, the linear trends will only approximate the ODS related contribution to*
*ozone changes.*

  p.8, l.215-216: This phrase is not entirely clear on whether or not you use the detrended proxy. This
choice is so central to this paper that it must be very clearly stated.  → *We added, "For these reasons, we do not detrend the proxy timeseries in this study".*

p.9, Fig. 1: chi^2 is the sum of "the squared differences median timeseries minus MLR" → *We changed to ".. sum square of differences between median and MLR timeseries' divided by ..."*

p.9, l.219: "MLR prediction after fitting" would be clearer than "MLR result from applying". → *better: "MLR timeseries derived from"*

p.9, l.220: To me, "after 1996" suggests 1996 is not included. What about replacing "after 1996" by "since 1996" throughout the manuscript? → *see earlier comment. It should be "after 1995" or "since*
*1996", similarly "before 1996" and "until 1995". We changed accordingly.*

    p.9, l.224: "recovery from reductions in ODS" would be more clear on the effect of ODS on ozone. →
*done*

    p.11, l.260: Replace "from applying" by "when applying"? → *leave it as is.*

p.11, l.260: It is somewhat unexpected to regress a "super" merged timeseries rather than average the trends from individual records. What is the rationale? Also, the sample size is just N=3, for 1979-1995, so
won't the "super"merge-then-regress method lead to more uncertainty in the MLR parameters than the regress-then-average approach? → *In Table 3 we present the trends of the median timeseries' as well as*
*the median and mean of the individual trends. The numbers are nearly the same.*

    p.12, Table 3 (caption): The periods in the caption are inconsistent with information in Figs 1 and 2. The
first trend period stops in 1995, the second starts in 1996. Hence, it should be 1979-1995 and 1996-2000. → *done*

p.12, Table 3: For each latitude belt, the occurrences of "mean/median trend >1996" should be >=1996, in order to be in line with Fig. 1 and 2. → *changed to t>1995*

p.12, Table 3: The error notation was confusing for me, I haven't seen this specific notation very often. For instance, what does -1.9(13)" mean? Is it -1.9+-0.13 or -1.9+-1.3 or ...? I find an explicit notation such
as "+0.4+-0.2" much more effective. I recommend using this throughout this table and also the manuscript. → *It is a common way to put uncertainties in the brackets, but I agree that this is not so*
*widely used in the atmospheric science community. In order to keep the table compact, we will remain with our notation.*

p.12, l.265: "One notable change from W18 is that the tropical trends during the ODS rising phase are now more negative (down to -1%/decade) while before they were mainly close to zero. This may be
caused by the additional proxy terms used in this study". The pre-1996 data have been available for a very long time now. Has this effect never been looked into before? If so, please refer to relevant work. -
→ *see our general comment in the beginning of the reply to reviewer #2*

    p.12, l.270: Please replace the "maybe" (conditional) by an "is" (certainty). Trend uncertainty scales with
n^{-3/2} (e.g., Weatherhead et al., 2000) so the eight more years in the recovery period already lead to ~45% smaller trend error. This seems not too far from the observed factor 2 reduction of the error in
Table 3 and Fig. 3. → *done*

    p.13, l.274: "The expected tropical recovery [...]". Estimated mid-lat NH recovery trends are too small
compared to EESC prediction as well. → *added "In the NH extratropics the expected ODS related*

*recovery is slightly higher than the observed trends, but also agree within the uncertainties of the observed trends."*

p.15, l.320: "NH total ozone has been steadily declining..." conflicts with the first phrase of this paragraph "stable ozone levels in NH since 2000". Please clarify the text. → *The stable levels refer to annual means at NH middle latitudes as shown in Figure 2 (added "middle latitude"), while the decline in*
*Figure 6 is shown for March only and also includes polar latitudes.*

p.15, l.324: "with larger springtime polar ozone losses"? → *done*

p.15, l.325: Remove "recent" from "A recent downward trend". Perhaps you meant that this was recently reported? Ball et al report a continuous decline since the 1980s, not a recent decline. → *done*

p.18, l.332: Quoted recovery trend value (11%/decade) conflicts with that in Figure 7 (12%/decade). Please correct. → *done*

p.19, Table 4: Same comment on error notation as in Table 3 (p.12). → *see earlier comment.*

p.19, l.367: The Gaudel paper is about differences between tropospheric ozone data records. So
probably not the best reference when the message is about consistency between tropo/strato/total ozone. → *We removed this sentence, as we did not mention tropospheric ozone at all in the paper.*
*When using annual mean zonal mean averages contribution of tropospheric ozone is likely to be small, but may become more important when looking at regional trends.*

5. Technical corrections

→ *all done*

p.1, l.10: Remove "on" from "[...] is indeed on slowly [...]".

p.1, l.12: Remove "in absolute numbers".

p.1, l.15: Add "-" to "chemistry-climate models".

p.2, l.30: Typo "stratosphere".

p.2, l.38: Remove "agreement" from "Montreal Protocol agreement".

p.3, l.75: Replace "in large part" by e.g. "largely".

p.3, l.79: Replace "Observations Zénithales" by "Observation Zénithale".

p.4, l.87: Replace "are processed using the same V8.7 retrieval algorithm" by e.g. "are retrieved using the same V8.7 algorithm".

p.4, l.108: Type "[...] shift to an equivalent [...]".

p.5, l.130-132: Double occurrence of ground-based. First one could be removed, e.g. "The WOUDC zonal
mean ...".

p.7, l.175: Add "." after "W18)".

220 p.7, l.189: Replace "there are not sufficient number of months" by e.g. "there are not enough months" or "there is not a sufficient number of months".

222 p.7, l.194: Replace MLR "equation" by MLR "model"?

   p.8, l.212: Remove "the possibility", as it is a bit redundant.

224 p.8, l.212: Replace "MLR results" by "MLR fit residuals" perhaps? This is a bit clearer as the MLR parameter estimates are MLR results as well.

226 p.9, l.218: "five bias-corrected" instead of "bias-corrected five".

   p.11, l.242-243: Maybe you forgot to remove the newline between paragraphs?

228 p.11, l.251: Add a "+" sign to the quoted values at start of this line.

   p.11, l.256: Remove ' after "timeseries".

230 p.11, l.261: Add "/decade " after +0.5%"

   p.12, Table 3 (caption): Remove "and" from caption "[...] in bold have an absolute [...]"

232 p.12, Table 3 (caption): Add "prediction" at the end of "and mod_ithe MLR".

   p.12, Table 3: Add $+$ to trend value $\geq$1996 for median time series near-global.

234 p.12, Table 3: The quoted r^2 value for WOUDC in 20S-20N band is single digit (0.7), should be double (0.70).

236 p.13, l.276: Remove "on" from "elucidate further on".

   p.13, l.285: Type "Fig. 4a" should be "Fig. 4".

238 p.15, Fig.5 (caption): There is a missing word in "Negative values an anti-correlation [...]".

   p.15, l.311: Add "s" to "chemical effect"?.

240 p.15, l.316: Add full stop at end of phrase.

   p.17, Fig.7 (caption): Capitalise "See".

242 p.18, l.331: "Earlier signs of ozone recovery have been", should be plural.

   p.18, l.331: Add "," in between "Now with".

244 p.18, l.332-333: "During September, the Antarctic ozone hole usually grows and [...]".

   p.18, l.340: Remove "as shown in Fig. 7". A bit redundant, you already referred to the figure in the
246 previous phrase.

   p.18, l.344: Replace "globally" by "global"?

248 p.18, l.352: Add "," in between "tropics recovery".

   p.18, l.354: Add "," in between "Arctic large".

p.19, l.363: "chemistry-climate models".

---

## Author Response (AR2)

Dear Michel,

As requested by the reviewer we removed the redundant sentences in the said paragraph. We also made very minor changes at other places in the manuscript (spelling, clarifications) which can be easily followed in the track changes of the attached manuscript.

Best wishes,

Mark

Reply to reviewer:

(1) *line 268 in the revised manuscript: You report trend numbers here, but the number for the SH regression with the standard MLR seems to be wrong. It is stated in the new Table 4 that the pre-1996 trend is-2.5%/decade, but here in the text it says -1.9%/decade. Please check these numbers in the table and the text.* → done
(2) *line 287-293 in the revised manuscript: This paragraph repeats many details of the paragraph before. Especially the last two sentences are identical. Please check the text and rephrase to avoid repetition.* → done

**Attachment:** manuscript with track changes

[revised manuscript text omitted]
$                                                                                          .